# Semi-Supervised Skin Lesion Segmentation under Dual Mask Ensemble with Feature Discrepancy Co-Training

**Thanh-Huy Nguyen**[*1]                                                      THANH-HUY_NGUYEN@ETU.U-BOURGOGNE.FR
**Hoang-Thien Nguyen**[*2,3]                                              N21DCCN080@STUDENT.PTITHCM.EDU.VN
**Xuan-Bach Nguyen**[2,4]                                                    BACH.NGUYENSPRING@HCMUT.EDU.VN
**Nguyen Lan Vi Vu**[2,4]                                                         VI.VUVIVU2203@HCMUT.EDU.VN
**Quang-Vinh Dinh**[2]                                                            VINH.DINHQUANG@AIVIETNAM.EDU.VN
**Fabrice Meriaudeau**[†1]                                                   FABRICE.MERIAUDEAU@U-BOURGOGNE.FR

[1] *Université Bourgogne Europe, CNRS, ICMUB UMR 6302, 21000 Dijon, France*

[2] *AI Vietnam Research Lab, 660000 Ninh Thuan, Vietnam*

[3] *Posts and Telecommunications Institute of Technology, Ho Chi Minh City, Vietnam*

[4] *Ho Chi Minh University of Technology, Vietnam*

**Editors:** Accepted for publication at MIDL 2025

## Abstract

Skin Lesion Segmentation with supportive Deep Learning has become essential in skin lesion analysis and skin cancer diagnosis. However, in the practical scenario of clinical implementation, there is a limitation in human-annotated labels for training data, which leads to poor performance in supervised training models. In this paper, we propose Dual Mask Ensemble (DME) based on a dual-branch co-training network, which aims to enforce two models to exploit information from different views. Specifically, we introduce a novel feature discrepancy loss trained with a cross-pseudo supervision strategy, which enhances model representation by encouraging the sub-networks to learn from distinct features, thereby mitigating feature collapse. Additionally, Dual Mask Ensemble training enables the sub-models to extract more meaningful information from unlabeled data by combining mask predictions. Experimental results demonstrate the effectiveness of our approach, achieving state-of-the-art performance across several metrics (Dice and Jaccard) on the ISIC2018 and HAM10000 datasets. Our code is available at https://github.com/antares0811/DME-FD.

## 1. Introduction

Lesion segmentation plays an important role in automated skin lesion analysis, as it facilitates the extraction of clinically relevant features such as lesion size, border irregularity, and contrast with the surrounding skin. While many successful machine learning models bypass explicit segmentation, studies have shown that these features contribute to improved lesion characterization and diagnosis (Marchetti et al., 2023). By accurately delineating the lesion, segmentation can enhance downstream tasks such as feature extraction and classification. However, manual annotation of skin lesion images is labor-intensive and prone to variability, making it difficult to produce large, accurately labeled datasets required for training robust models. These challenges underscore the importance of semi-supervised

---

[*] Contributed equally

[†] Corresponding Author

learning approaches, which leverage both labeled and unlabeled data to reduce dependence on extensively labeled datasets while improving real-world model generalization deployment.

In recent years, semi-supervised learning (SSL) techniques have gained significant attention for training models with limited pixel-wise annotated data and a larger set of unlabeled data. Among these, pseudo-labeling methods (Yang et al., 2022; Mendel et al., 2020) are widely used. However, they often face challenges related to confirmation bias (Yang et al., 2022), where incorrect pseudo-labels reinforce errors during training, leading to performance degradation due to training instability. Consistency regularization-based methods (Sohn et al., 2020; Yang et al., 2023) generate predictions from weakly perturbed inputs to create pseudo-labels but still remain vulnerable to confirmation bias issues.

Conversely, co-training allows different sub-networks to infer the same instance from various perspectives and transfer knowledge from one view to another through pseudo-labeling. Co-training, in particular, leverages multi-view references to improve the model's perception and increase the reliability of the pseudo-labels generated (Qiao et al., 2018). Cross-pseudo supervision (CPS) (Chen et al., 2021) enforces consistency between the outputs of two networks by using cross-network pseudo-labels. CCVC (Wang et al., 2023) proposes a cross-view consistency strategy that pushes the feature extractor outputs of two networks apart, enabling the sub-networks to learn richer semantic information from conflicting predictions. (Zeng et al., 2024) employs a single-encoder dual-decoder architecture, where differential decoder features are then served as feedback signals to the encoder.

To design an effective method that prevents sub-networks from collapsing into similar, ineffective representations, we revisit the dual-branch networks (Chen et al., 2021) and extend it with a proposal of a Dual Mask Ensemble (DME) for semi-supervised segmentation. Unlike CPS, our method leverages not only the information from the opposing subnet but also its own generated mask. This self-generated mask is combined with the opponent's predicted mask to guide the model during backpropagation. Specifically, we first introduce the Dual Mask Ensemble, a mask combination technique designed to enable the model to extract additional information from unlabeled data, thereby enhancing its ability to produce precise and reliable predictions. Similar to (Wang et al., 2023; Zeng et al., 2025), to prevent the sub-networks from collapsing into similar representations, we propose a new feature discrepancy loss that encourages the models to extract distinct features, thus diversifying their representation space. However, (Wang et al., 2023) relies on conflict-based consistency but lacks an explicit mechanism to address low-confidence predictions. In contrast, our advanced DME module adaptively combines predictions from dual sub-networks based on their confidence and consistency, resulting in more stable pseudo-labels and better generalization. Furthermore, unlike (Zeng et al., 2025), which focuses on decoder-level discrepancy learning, our Feature Discrepancy module emphasizes learning from representation-level discrepancies between sub-networks, enhancing the diversity and complementarity of their predictions. Our contribution can be summarized as follows:

- We introduce the Dual Mask Ensemble, integrated with a dual-branch co-training framework, to enhance the model's ability to generate more reliable predictions.

- We propose a novel feature discrepancy loss that promotes the extraction of distinct features, effectively diversifying the model's representation space.

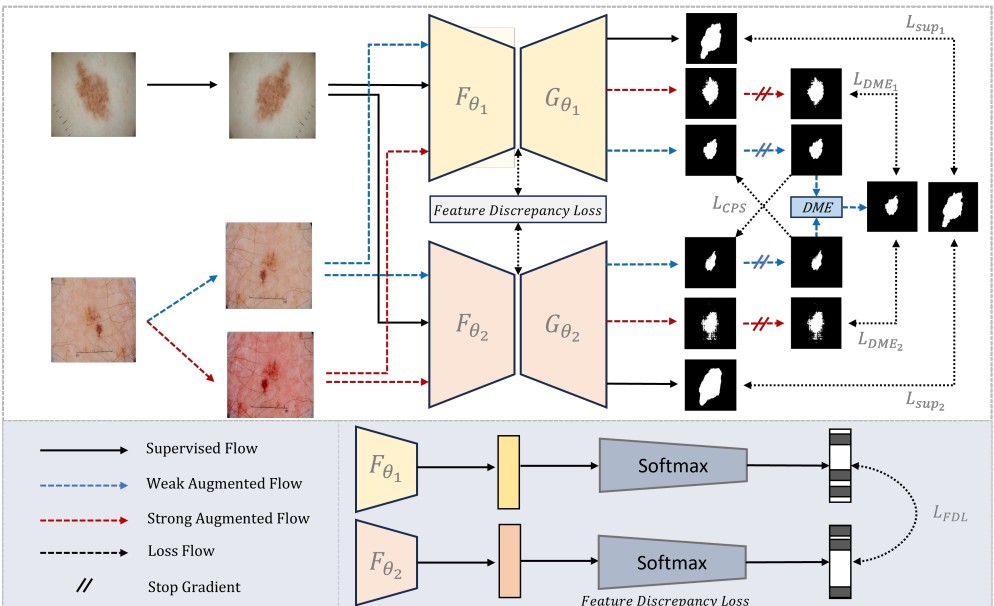

Figure 1: Overview of our proposed method. The Dual Mask Ensemble module combines masks predicted from weakly augmented inputs into a reliable mask and computes the DME loss with those predicted from strongly augmented inputs. The feature discrepancy loss is applied to features from both sub-networks' encoder outputs.

- Extensive experiments with our method on the ISIC2018 (Codella et al., 2018) and HAM10000 (Tschandl et al., 2018, 2020) datasets show state-of-the-art performance, demonstrating our robustness in the semi-supervised skin segmentation task.

## 2. Methodology

Given a set of label images $D_l = \{(x^l, y^l)\}$ along with unlabeled images $D_u = \{x^u\}$. The main objective is to leverage information from the unlabeled set through two distinct training flows: Cross-Pseudo Supervision training (Chen et al., 2021) and Dual Mask Ensemble training (2.1). However, using both flows may lead to model collapse, where the predictions of the two models become identical. To address this, we propose a feature discrepancy loss (2.2) to preserve the diversity between the model views. A brief overview of our pipeline is provided in 2.3 and illustrated in Figure 1.

### 2.1. Dual Mask Ensemble

To fully exploit the information from unlabeled data, we adopt a weak-to-strong paradigm to help each model understand the semantic meaning of images by themselves. Let $A_w$ and $A_s$ denote weak and strong augmentations, respectively. Weak augmentation involves Random Flipping, while strong augmentations include Gaussian Noise, Brightness Contrast, and Color Jittering.

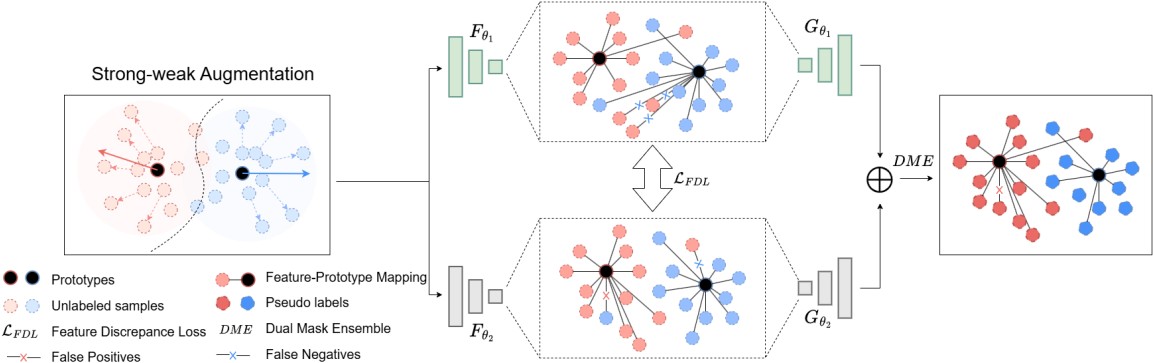

Figure 2: Visualization of the combination of Strong-Weak Augmentation, Feature Discrepancy Loss, and the Dual Mask Ensemble module from a feature perspective.

Firstly, the unlabeled input $x^u$ is transformed into weak $(X_w)$ and strong $(X_s)$ augmented versions as $X_w = A_w(x^u)$ and $X_s = A_s(A_w(x^u))$. Then, the transformed inputs are fed into each model to obtain the confidence maps:

$$P_1^W = g_1(f_1(X_w)), P_1^S = g_1(f_1(X_s)), \qquad P_2^W = g_2(f_2(X_w)), P_2^S = g_2(f_2(X_s)). \quad (1)$$

Finally, we compute the loss between them. Both one-hot label maps of the weak ones are integrated to guide the stronger ones. However, raw predictions from weakly augmented images may contain noise, which can degrade model performance. To mitigate this, a fixed confidence threshold $\tau$ is applied:

$$Y_1^W = \mathbb{1}\left(\max(p_i) \geq \tau\right) \arg\max_c p_i(y = c \mid P_1^W), \qquad (2)$$

$$Y_2^W = \mathbb{1}\left(\max(p_i) \geq \tau\right) \arg\max_c p_i(y = c \mid P_2^W). \qquad (3)$$

Here, $\tau$ serves to separate object pixels from background pixels, $Y_1^W$ and $Y_2^W$ are the pseudo-label masks from the two models, which are then combined using the summation (OR) operation:

$$\hat{Y}^W = Y_1^W \oplus Y_2^W \qquad (4)$$

The loss for the Dual Mask Ensemble (DME), $L_{DME}$ is defined as:

$$L_{DME} = L_{bce,dice}(P_1^S, \hat{Y}^W) \quad + L_{bce,dice}(P_2^S, \hat{Y}^W) \qquad (5)$$

### 2.2. Feature Discrepancy Loss

The combination of both cross-supervision loss and DME loss can lead to model collapse, where all models produce identical predictions for a sample (Wu and Cui, 2024). To prevent this issue, we propose a feature discrepancy loss that ensures diversity in the model predictions by maintaining differences in the representation space. The feature discrepancy loss $(L_{dis})$, indicated in Figure 2, is defined as:

$$L_{dis}(f_1, f_2) = \frac{1}{D(f_1, f_2) + \epsilon} \qquad (6)$$

where $\epsilon = 1e^{-6}$ prevents division by zero, $f_1$ and $f_2$ are the features from two models, and $D$ represents the Manhattan distance function.

We first extract the feature representations from the model encoder's output. $F_1^{sup}$ and $F_1^w$ are the features of supervised and weakly augmented samples from the first model, while $F_2^{sup}$ and $F_2^w$ are the corresponding features for the second model. Next, we normalize the feature values using the $Softmax$ function:

$$F_1^{sup}, F_1^w = Softmax(f_1([x^l, X_w])), \qquad F_2^{sup}, F_2^w = Softmax(f_2([x^l, X_w])). \quad (7)$$

Finally, the feature discrepancy loss is applied to both the supervised and weakly augmented features:

$$L_{FDL} = \frac{1}{2}(L_{dis}(F_1^{sup}, F_2^{sup}) + L_{dis}(F_1^w, F_2^w)) \quad (8)$$

### 2.3. Overall framework

Overall, the final objective loss is written as:

$$L = L_{sup} + \alpha(L_{cps} + L_{DME}) + \beta L_{FDL} \quad (9)$$

where $\alpha$ is Consistency Warm-up in (Laine and Aila, 2017). Although using the feature discrepancy loss can increase the model's diversity between different views, it could harm the model by not getting the convergent point in the last epochs. To avoid this behavior, $\beta = 10^{-t/(T*0.25)}$ is added as a decay for $L_{FDL}$, where t is the current epoch and T is the maximum number of epochs.

## 3. Experiments

### 3.1. Experimental Settings

We evaluated our proposed methods on two publicly available datasets dedicated to the skin lesion segmentation task. The number of labeled samples is selected by 1%, 2%, 4% of the total training samples, and the rest were used as unlabeled data. We also adopted 5-fold cross-validation to measure model performance.

**ISIC-2018:** The ISIC-2018 (Codella et al., 2018) dataset contains 3694 images with labeled masks. We used 2955 samples for training and 739 samples for evaluating the performance.

**HAM10000:** The HAM10000 (Tschandl et al., 2018, 2020) dataset consists of 10015 samples, partitioned into 8012 samples for training and 2003 samples for validation.

### 3.2. Implementation Details

The proposed method was implemented with PyTorch and trained on a single NVIDIA RTX A6000 card with 48 GB of memory. SwinUnet (Cao et al., 2023) is utilized as our main model architecture. We use the AdamW optimizer with an initial learning rate of $1 \times 10^{-4}$, a linear decay scheduler whose step size is 50 and decay factor $\gamma = 0.5$. The input images were resized to $224 \times 224$. The batch size was set to 8 for ISIC-2018 and 24 for HAM10000. The model was trained for 80 epochs. In the augmentation stages, we adopted Random Flipping for weak augmentation, while Random Color Distortion, Color Jitter,

Table 1: Quantitative results on the ISIC-2018 under two labeled ratio configurations. **L** and **U** are the training ratios of labeled and unlabeled sets, respectively.

| Method | Ratio (%) | | Metrics | | | |
|---|---|---|---|---|---|---|
| | L | U | Dice (%) | JC (%) | PRE (%) | ACC (%) |
| | 2 | - | 74.65 ±2.92 | 60.81 ±2.99 | 76.09 ±7.54 | 89.22 ±1.52 |
| SupOnly | 4 | - | 77.23 ±0.48 | 65.35 ±0.56 | 80.28 ±1.60 | 90.78 ±0.30 |
| | 100 | - | 87.66 ±0.93 | 78.49 ±1.38 | 88.35 ±1.12 | 94.86 ±0.31 |
| PseudoSeg | | | 76.34 ±3.88 | 64.29 ±5.03 | 81.92 ±4.28 | 90.30 ±2.01 |
| CCT | | | 75.11 ±4.10 | 62.99 ±5.79 | 81.01 ±2.40 | 89.23 ±2.93 |
| CPS | 1 | 99 | 76.59 ±3.86 | 64.31 ±3.81 | 81.98 ±3.05 | 90.01 ±1.40 |
| GTA-Seg | | | 75.67 ±4.49 | 63.87 ±4.18 | 78.44 ±6.01 | 89.55 ±1.39 |
| UniMatch | | | 77.16 ±3.16 | 65.05 ±4.45 | **82.43** ±5.17 | 90.41 ±2.19 |
| **Ours** | | | **78.63** ±2.32 | **66.28** ±3.98 | 82.12 ±2.87 | **90.86** ±1.66 |
| PseudoSeg | | | 79.76 ±2.11 | 67.16 ±2.77 | **84.56** ±2.29 | 91.67 ±1.12 |
| CCT | | | 78.66 ±2.02 | 65.80 ±2.63 | 81.84 ±1.85 | 91.28 ±1.02 |
| CPS | 2 | 98 | 79.61 ±1.66 | 67.04 ±2.28 | 82.24 ±2.81 | 91.56 ±0.86 |
| GTA-Seg | | | 77.33 ±2.20 | 64.21 ±2.59 | 76.65 ±5.66 | 90.30 ±0.73 |
| UniMatch | | | 80.03 ±2.04 | 67.55 ±2.71 | 83.30 ±3.87 | 91.74 ±1.00 |
| **Ours** | | | **80.07** ±1.75 | **67.62** ±2.37 | 82.59 ±1.52 | **91.75** ±0.98 |
| PseudoSeg | | | 81.77 ±0.66 | 71.18 ±1.03 | **85.23** ±2.47 | 92.72 ±0.30 |
| CCT | | | 80.96 ±1.11 | 68.95 ±1.41 | 83.37 ±0.83 | 92.22 ±0.55 |
| CPS | 4 | 96 | 80.89 ±0.91 | 70.31 ±1.07 | 83.90 ±2.26 | 92.29 ±0.28 |
| GTA-Seg | | | 80.83 ±0.80 | 70.03 ±1.07 | 83.30 ±2.45 | 91.96 ±0.84 |
| UniMatch | | | 81.41 ±1.22 | 69.46 ±1.58 | 84.51 ±2.05 | 92.43 ±0.77 |
| **Ours** | | | **82.06** ±0.69 | **71.54** ±1.04 | 84.81 ±1.55 | **92.83** ±0.40 |
| PseudoSeg | | | 83.96 ±0.86 | **73.08** ±1.27 | 85.98 ±2.62 | **93.48** ±0.25 |
| CCT | | | 83.65 ±0.93 | 72.58 ±1.36 | 85.32 ±2.40 | 93.24 ±0.25 |
| CPS | 8 | 92 | 83.75 ±0.74 | 72.77 ±1.14 | 85.04 ±1.45 | 93.34 ±0.13 |
| GTA-Seg | | | 83.65 ±0.98 | 72.62 ±1.51 | 83.98 ±1.38 | 93.21 ±0.50 |
| UniMatch | | | 83.90 ±0.56 | 72.89 ±0.80 | 84.64 ±2.20 | 93.30 ±0.06 |
| **Ours** | | | **84.00** ±0.31 | 73.06 ±0.52 | **86.58** ±0.52 | 93.44 ±0.24 |

and Gaussian Noise were implemented for strong augmentation. The confidence threshold $\tau$ was set to 0.85. We evaluated performance using mean Dice similarity coefficient (Dice), Jaccard coefficient (JC), precision (PRE), and accuracy (ACC).

## 3.3. Comparison With Existing Methods

Table 2: Quantitative results on the HAM10000 under two labeled ratio configurations.

| Method | Ratio (%) | | Metrics | | | |
|---|---|---|---|---|---|---|
| | L | U | Dice (%) | JC (%) | PRE (%) | ACC (%) |
| | 2 | - | 88.15 ±0.21 | 78.90 ±0.31 | 88.12 ±0.42 | 93.73 ±0.07 |
| SupOnly | 4 | - | 89.59 ±0.07 | 81.24 ±0.12 | 90.83 ±1.17 | 94.56 ±0.09 |
| | 100 | - | 93.54 ±0.25 | 87.92 ±0.42 | 93.89 ±0.57 | 96.58 ±0.16 |
| PseudoSeg | | | 90.02 ±0.17 | 81.94 ±0.28 | 92.11 ±1.26 | 94.81 ±0.18 |
| CCT | | | 89.93 ±0.10 | 81.79 ±0.15 | 91.55 ±0.96 | 94.75 ±0.11 |
| CPS | 2 | 98 | 89.94 ±0.14 | 81.81 ±0.23 | 92.21±0.77 | 94.78 ±0.15 |
| GTA-Seg | | | 89.55 ±0.32 | 81.17 ±0.54 | 90.39 ±0.10 | 94.48 ±0.21 |
| UniMatch | | | 89.66 ±0.15 | 81.35 ±0.26 | 91.68 ±0.60 | 94.62 ±0.20 |
| **Ours** | | | **90.45** ±0.17 | **82.65** ±0.27 | **92.40** ±1.02 | **95.04** ±0.20 |
| PseudoSeg | | | 90.97 ±0.39 | 83.21 ±0.64 | **92.72** ±1.19 | 95.20 ±0.28 |
| CCT | | | 90.64 ±0.53 | 82.97 ±0.86 | 92.43 ±0.15 | 95.12 ±0.25 |
| CPS | 4 | 96 | 90.76 ±0.51 | 83.17 ±0.84 | 92.56 ±0.46 | 95.16 ±0.29 |
| GTA-Seg | | | 90.86 ±0.19 | 83.34 ±0.31 | 92.18 ±0.54 | 95.21 ±0.12 |
| UniMatch | | | 90.32 ±0.44 | 82.43 ±0.73 | 91.96 ±1.21 | 94.95 ±0.30 |
| **Ours** | | | **91.13** ±0.30 | **83.79** ±0.50 | 92.36 ±0.30 | **95.34** ±0.19 |

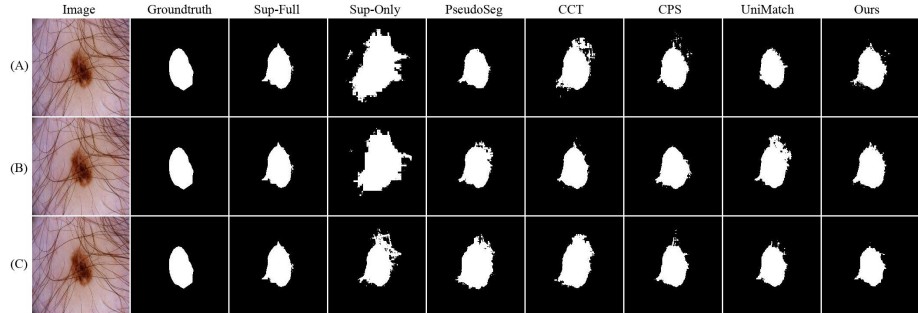

Figure 3: Visualization of semi-supervised model performance on ISIC2018 dataset under various supervised training sample ratio: A: 2%; B: 4%; C: 8%

### 3.3.1. Quantitative Comparison

Our proposed framework is fairly compared with PseudoSeg, CCT, CPS, and UniMatch on ISIC-2018 and HAM10000. Quantitative results are detailed in Table 1 and Table 2. A supervised baseline using only labeled data ("SupOnly") is also evaluated. All methods employ the same data augmentation, training strategies, and backbones to ensure fair comparisons.

**Segmentation Results on ISIC-2018:** Table 1 compares our method with other semi-supervised segmentation frameworks on the ISIC-2018 dataset. With a setting of limited 2% labeled data (59 labeled and 1896 unlabeled samples), our approach achieves notable improvements in both the Dice score (80.07%) and the Jaccard coefficient (67.62%), outperforming all competing methods. When the ratio of labeled data increases to 4% (118 samples), the Dice and Jaccard scores further improve to 82.06% and 71.54%, maintaining the leading position. With an 8% labeled dataset (236 labeled and 2791 unlabeled samples), our method achieves the highest Dice score (84.12%) and ranks second in the Jaccard coefficient (73.24%), slightly below the full-supervised baseline, while surpassing state-of-the-art methods like UniMatch and CPS.

**Segmentation Results on HAM10000:** Table 2 displays a comparison of our performance with other semi-supervised segmentation frameworks on the HAM10000 dataset. Provided a limited set of 2% labeled data (160 labeled and 7852 unlabeled images), our approach shows a marked improvement in both Dice score and Jaccard coefficient, achieving 90.45% and 82.65%, respectively. With 4% (320) labeled images, our method achieves the highest performance, with a Dice score of 91.13% and a Jaccard coefficient of 83.79%.

### 3.3.2. Qualitative Comparison

Figs. 3 and 4 visually compare the proposed method with existing approaches, alongside the original images, ground-truth labels, and full-supervised predictions for a detailed assessment. Our method clearly delivers smoother predictions with fewer blending pixels compared to other methods. We also visualized the effectiveness of our method on different types of lesions and cross-domain scenarios between HAM10000 and ISIC2018. The detailed figures are provided in the Appendix.

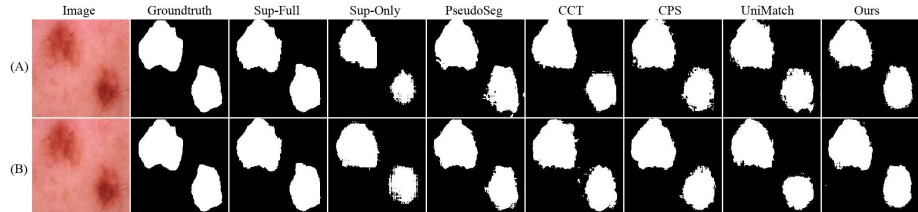

Figure 4: Visualization of semi-supervised model performance on the HAM10000 dataset under various supervised training sample ratio: A: 2%; B: 4%

## 3.4. Ablation Study

Table 3: Results on Mask Refinement Training with 4% Labeled Samples in two datasets

| Method | ISIC-2018 | | HAM10000 | |
|---|---|---|---|---|
| | Dice (%) | JC (%) | Dice (%) | JC (%) |
| Intersect | 81.41 ±0.68 | 70.82 ±0.99 | 91.00 ±0.36 | 83.56 ±0.60 |
| Union | **82.06** ±0.69 | **71.54** ±1.04 | **91.13** ±0.30 | **83.79** ±0.50 |
| Self-teaching | 81.39 ±1.10 | 70.73 ±1.49 | 91.08 ±0.36 | 83.71 ±0.59 |
| Cross-view teaching | 81.49 ±0.97 | 70.89 ±1.38 | 91.07 ±0.42 | 83.70 ±0.70 |

### 3.4.1. Mask Refinement Mechanism

Table 3 compares four different approaches of mask integration for skin lesion segmentation on the ISIC2018 and HAM10000 datasets, using 4% of labeled samples. The investigated approaches include Intersect, Union, Self-Teaching, and Cross-View Teaching.

**Intersect Method** employs a multiplication (AND) operation for mask integration, aiming to retain only the overlapping regions between different predictions. The performance, shown in Table 3, indicates that the strict intersection strategy can effectively filter out noisy predictions but risks discarding valuable information, leading to lower scores compared to other methods.

**Union Method** applies a summation (OR) operation to combine masks, encompassing all possible regions covered by different predictions. This method, adopted as our current approach, exhibits superior performance, particularly on the ISIC2018 dataset, with a Dice coefficient of 82.06% ± 0.69 and a JC of 71.54% ± 1.04. Similarly, in the HAM10000 dataset, the Union approach continues to deliver top performance with a Dice of 91.13% ± 0.30 and

Table 4: Ablation studies of our framework with 4% labeled samples on ISIC2018

| Sup | CPS | DME | FDL | Dice (%) | JC (%) |
|---|---|---|---|---|---|
| ✓ | | | | 77.23 ±0.48 | 65.35 ±0.56 |
| ✓ | ✓ | | | 79.61 ±1.66 | 67.04 ±2.28 |
| ✓ | ✓ | ✓ | | 81.67 ±0.88 | 71.13 ±1.22 |
| ✓ | ✓ | ✓ | ✓ | **82.06** ±0.69 | **71.54** ±1.04 |

JC of 83.79% ± 0.50. These results underline that the Union method effectively integrates multiple predictions, with complete ROI capture thanks to a comprehensive mask.

**Self-Teaching Method** (Zou et al., 2021) uses the weaker version of a pseudo mask to guide its own refinement towards a stronger version. While the Self-Teaching yields slightly lower scores than the Union, it demonstrates competitive performance, especially in challenging cases where weak pseudo masks iteratively refine to deliver accurate predictions.

**Cross-View Teaching Method** (Ngo et al., 2024) involves cross-guidance, where a weak pseudo mask supervises predictions of stronger augmented images from the opposite model. This approach achieves performance comparable to the Self-Teaching. However, the added complexity of Cross-View Teaching does not consistently outperform the Union.

### 3.4.2. ANALYSIS ON COMPONENT EFFECTIVENESS

Our method incorporates several key components: a CPS module, a Dual Mask Ensemble (DME) module, and a feature discrepancy strategy. Table 4 investigates the individual contributions of these components on the ISIC2018 dataset with 4% supervised samples.

Applying cross-pseudo supervision loss ($L_{cps}$) improves Dice and JC metrics by over 2% and 1.7%, showing its effectiveness despite some correlation between sub-net views. Leveraging the DME module ($L_{DME}$) further boosts Dice by 2% and Jaccard by 4%. Finally, adding feature discrepancy loss ($L_{FDL}$) increases both metrics by 0.4%, enabling sub-nets to learn from orthogonal views and outperforming state-of-the-art methods.

### 3.4.3. ANALYSIS ON FEATURE LOSS SELECTION

| Method | L | U | Dice (%) | JC (%) | PRE (%) | ACC (%) |
|---|---|---|---|---|---|---|
| CCVC | 4% | 96% | 81.61 ±1.42 | 71.01 ±1.97 | **85.49** ±0.66 | 92.72 ±0.74 |
| **FDL** | | | **82.06** ±0.69 | **71.54** ±1.04 | 84.81 ±1.55 | **92.83** ±0.40 |
| CCVC | 8% | 92% | 83.67 ±0.59 | 72.66 ±0.80 | 84.80 ±1.21 | 93.33 ±0.06 |
| **FDL** | | | **84.00** ±0.31 | **73.06** ±0.52 | **86.58** ±0.52 | **93.44** ±0.24 |

Table 5: Feature loss design comparison on the ISIC-2018

We compared our method with the most relevant approach on feature correlation between two networks - CCVC (Wang et al., 2023). In contrast, FDL leverages modified Manhattan distance-based to enforce the difference between two feature representations. In Table 5, our loss design achieved a clear improvement compared to CCVC in most metrics. A deeper analysis of FDL is provided in Appendix A, Table 6 and Table 7 specifically.

## 4. Conclusion

In this work, we present a semi-supervised method based on a co-training framework for skin lesion segmentation. We have introduced the Dual Mask Ensemble module to enhance the model's ability to learn meaningful information from unlabeled data. Additionally, we demonstrate that our proposed feature discrepancy loss boosts model performance by encouraging distinct feature extraction, which avoids the collapse and diversifies the representation space of models, thus reducing the confirmation bias problem. Extensive experiments on benchmark datasets validate the robustness of the proposed approach.

## 5. Acknowledgement

We would like to thank the Graduate School INTHERAPI for its financial support.

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

## Appendix A. Analysis on Feature Discrepancy Loss

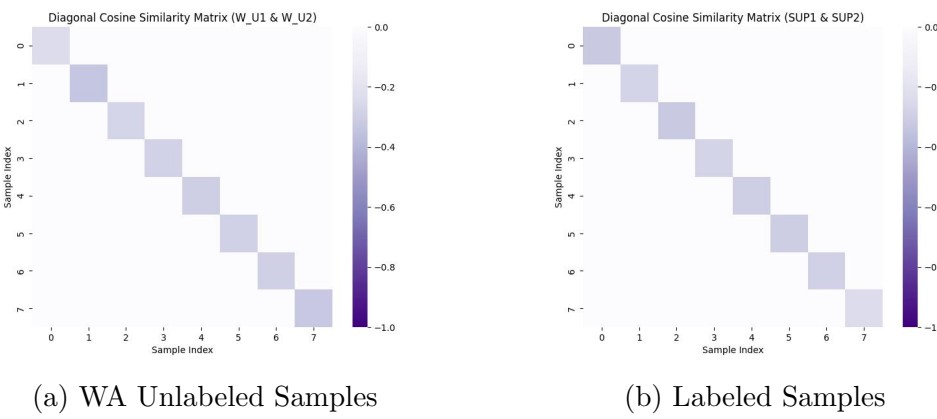

| (a) WA Unlabeled Samples | (b) Labeled Samples |

Table 6: Sample-wise Feature Correlation Using Cosine Similarity betweem both branches in dual-network, where WA denotes the weak augmentation.

**Impact of Feature Discrepancy Loss** To check whether the two parallel models utilize complementary or less-correlated features, we visualized the correlation between features of each sample among branches using cosine similarity. Following the Table 6, the diagonal elements being close to zero indicate that the feature representations from the two branches exhibit low similarity, suggesting that they capture distinct aspects of the data. Moreover, the observed differences between the labeled and unlabeled samples, where the labeled samples show slightly stronger decorrelation, support the idea that the feature discrepancy loss indeed encourages diverse feature learning. This addresses the concern about merely having shifted versions of similar feature vectors — if that were happening, we would expect more consistent and higher correlation patterns across the matrices. Instead, the observed variation and consistently low cosine similarity demonstrate that the models learn complementary and non-redundant features.

**Comparison to discrepancy loss of CCVC:** Table 7 provides valuable insight into the distinct behavior of our proposed discrepancy loss compared to the CCVC discrepancy loss. The diagonal cosine similarity values being close to zero reflect the degree of feature discrepancy between the two branches of the network. In the first figure ((a) - CCVC), the consistently strong negative correlations along the diagonal indicate a more rigid and potentially less adaptive discrepancy mechanism. In contrast, our method (second figure (b)) shows a more nuanced and flexible distribution of similarity values — this suggests that our approach captures a richer diversity in feature representations, likely leading to more robust and generalizable model performance. This highlights the advantage of our discrepancy loss in fostering complementary and well-differentiated feature learning between the branches.

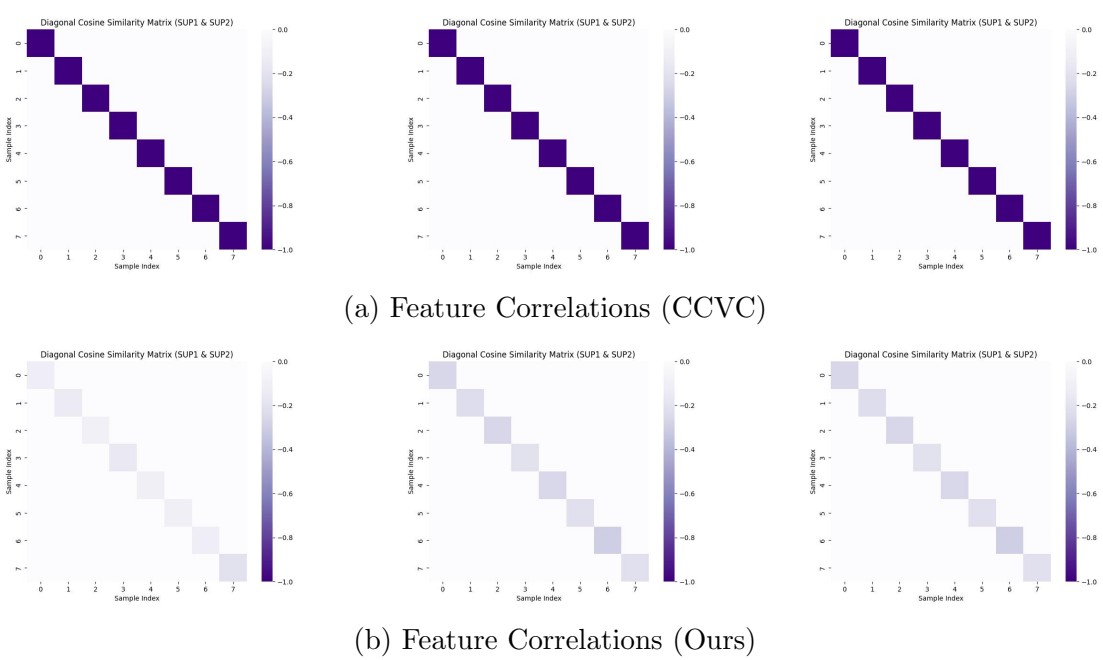

(a) Feature Correlations (CCVC)

(b) Feature Correlations (Ours)

Table 7: Visualization of feature correlation of each sample. **Left**, **Middle**, **Right** represent for the **early**, **middle** and **late** iteration.

## Appendix B. Rare skin lesion types

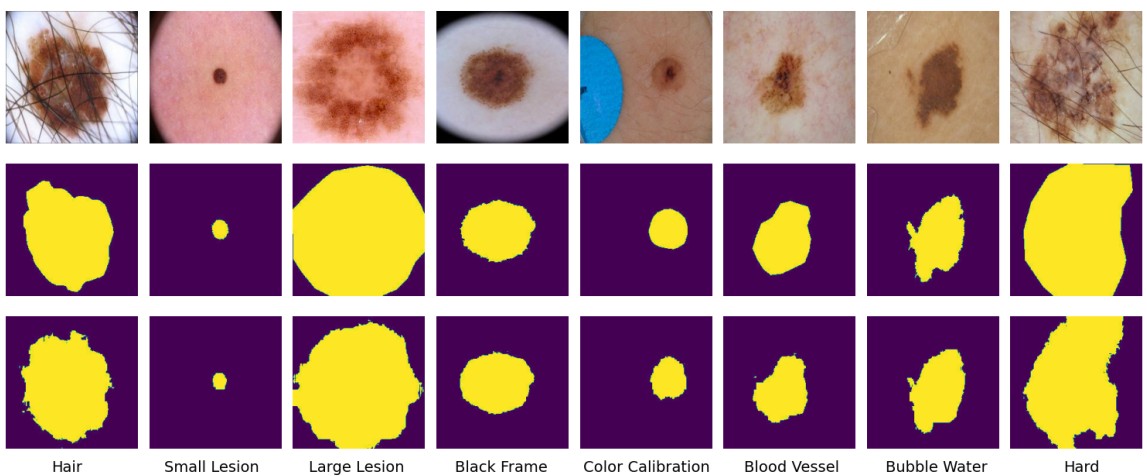

Figure 5: The visualization of different types of skin on ISIC dataset

As seen in the provided figure 5, the first row shows the original images with various challenges — including hair artifacts, small and large lesions, black frames, color calibra-

tion marks, blood vessels, water bubbles, and overall complex or 'hard' cases. The second row illustrates the ground truth segmentations, while the third row shows our model's predictions. Despite these difficult conditions, our approach consistently captures the lesion areas with high fidelity, maintaining accurate boundaries and minimizing false positives and negatives. Notably, even in cases with heavy occlusion (like hair) and small or irregularly shaped lesions, our method remains resilient, demonstrating its generalization ability across diverse and challenging data distributions. This highlights the robustness and effectiveness of our approach in real-world clinical scenarios.

## Appendix C.  Cross-Domain Performance Evaluation

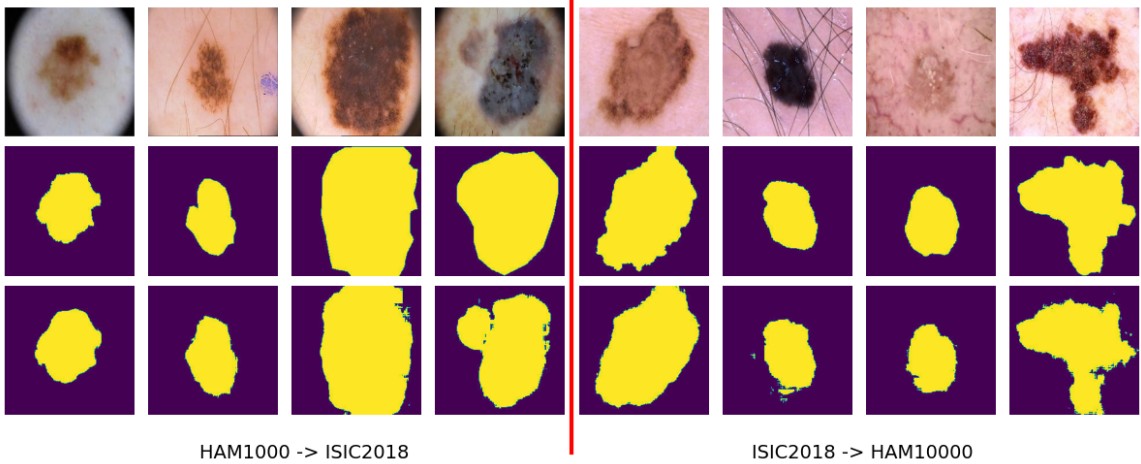

Figure 6: The visualization of prediction with cross-domain between ISIC2018 and HAM10000. The top, second, and bottom rows indicate the images, groundtruth images, and predictions of models, respectively.

We perform cross-domain evaluation by training on one dataset and evaluating on the other. Specifically, we use the best-performing model weights from each dataset (ISIC2018 and HAM10000) and test them on the other dataset. The visual Figure 6 shows that while the segmentation performance generally transfers well, there are noticeable differences in mask quality, particularly in shape and boundary accuracy, indicating domain shifts between the datasets. This cross-domain evaluation highlights the model's robustness and its limitations when adapting to unseen data distributions.

