# OpenReview forum: "Semi-Supervised Skin Lesion Segmentation under Dual Mask Ensemble with Feature Discrepancy Co-Training"
_MIDL.io/2025/Conference — MIDL 2025 Poster_

### Official Review · Reviewer_pE22 · 2025-02-18

**Confidence:** 5
**Preliminary Rating:** 4
**Recommendation:** Poster
**Final Rating:** 4

**Summary:**

The study presents an innovative framework for skin lesion segmentation known as Dual Mask Ensemble (DME), specifically addressing the issue of limited human-annotated training data prevalent in clinical practice. The proposed methodology utilizes a dual-branch co-training network to facilitate information exchange between two models from distinct perspectives. Central to this framework is introducing a novel feature discrepancy loss, which employs a cross-pseudo supervision strategy to encourage sub-networks to capture diverse feature representations, thus reducing the risk of feature collapse. Empirical results validate the efficacy of the proposed method, demonstrating performance on various metrics, including Dice and Jaccard coefficients, across two distinct datasets.

**Strengths:**

The presented technical innovations are targeted to solve particular issues related to the aimed medical image processing problem.
The design of the new loss functions are well justified
Comparative studies are presented
Ablation studies are presented

**Weaknesses:**

Results on the ISIC dataset are not statistically significant compared to the SSL methods.

The supervised model is not clear. Is it a single Swin model (just one of the branches)?

The presented model uses two Swin Unets together, so it has double the capacity of the supervised model (assuming the supervised model is a single Swin UNET).

It is not clear how the labeled vs. unlabeled samples were selected in the training dataset. Did the authors conduct cross-validation on this selection?

**Detailed Comments:**

The opening sentence of the paper is not 100% correct, as the majority of the ML models  (including the most successful ones) in the literature do not rely on any lesion segmentation. That being said, lesion segmentation is proven to be an important component in lesion diagnosis, as the lesion size, border jaggedness, and contrast between the lesion and the background are proven to be important parameters in diagnosis (https://doi.org/10.1111/jdv.18924)

In the presented setting, the SSL model has nearly twice as many trainable parameters (since it uses two parallel networks) compared to the supervised model. Does this impact the results? Did the authors try to increase the capacity of the supervised model by increasing its trainable parameters (e.g., widening the intermediate layers)?

Through the feature discrepancy loss, the authors aim to make the two parallel models as uncorrelated as possible. However, these two models are still attempting to solve the same problem, meaning that they effectively aim to find two different approaches to address the same issue. It remains unclear whether this method is an efficient way to tackle the problem. Although feature discrepancy loss encourages the model to find (utilize) different feature vectors, this does not necessarily imply that they are not extracting similar information. Ultimately, both models can converge to very similar parameters, where the model parameters may only shift in position (the order of filters in the layers). This can lead to the minimization of feature discrepancy loss simply by adjusting the positions of the features.

**Justification Of The Final Rating:**

I want to thank the authors for their replies. Based on their responses, I would still like to keep my score as it is. Even if the authors provided response to all my questions and comments, the responses do not change the impact and novelty of the research that are presented.

**Justification Of The Preliminary Rating:**

The paper is well-written and organized. It is easy to follow and read.
The authors base their technical novelties on open medical problems
The experiments are designed well to test most of the aims. There are some questions about the efficacy of the feature discrepancy loss (raised in the questions and detailed comments sections)

**Questions To Address In The Rebuttal:**

In the presented setting, the SSL model has nearly twice as many trainable parameters (since it uses two parallel networks) compared to the supervised model. Does this impact the results? Did the authors try to increase the capacity of the supervised model by increasing its trainable parameters (e.g., widening the intermediate layers)?

Is it possible to check if the two parallel models utilize complementary or less-correlated features? The aim of the feature discrepancy loss is that, however, is it possible to check if it serves the purpose? The feature discrepancy loss can be minimized by just having two shifted versions of similar feature vectors, which results from shuffling the order of the filters in the intermediate levels.

**Special Issue:**

No

---

> ### Author Response · Authors · 2025-03-08
>
> We are grateful to the reviewer for their thoughtful feedback and valuable suggestions. In the following, we carefully address each concern and aim to clarify any misunderstandings.
>
> $\textbf{C1:}$ "Results on the ISIC dataset are not statistically significant compared to the SSL methods."
>
> It is true that we did not achieve impressive results from the proposed method compared to UniMatch, especially on 8\% setting of ISIC-2018. However, our extended experimental setting on ISIC-2018 reveals a more promising performance - the smaller the amount of labeled samples, the more robust we achieved with our method. Please refer to the main ISIC-2018 quantitative table.
>
> $\textbf{Q2:}$ "The supervised model is not clear. Is it a single Swin model (just one of the branches)?"
>
> The supervised model is a single SwinUnet, identical to one of the branches in our proposed method.
>
> $\textbf{Q3:}$ "It is not clear how the labeled vs. unlabeled samples were selected in the training dataset. Did the authors conduct cross-validation on this selection?"
>
> We used 5-fold cross-validation, where one fold is selected as the validation set, and the remaining four folds are split into labeled and unlabeled sets according to the specified ratio.
>
> $\textbf{Q4:}$ "In the presented setting, the SSL model has nearly twice as many trainable parameters (since it uses two parallel networks) compared to the supervised model. Does this impact the results? Did the authors try to increase the capacity of the supervised model by increasing its trainable parameters (e.g., widening the intermediate layers)?"
>
> While it’s true that our SSL baseline uses two parallel networks during training, it’s important to clarify that during inference, we only use a single SwinUnet — exactly like the supervised-only (SupOnly) setting. This ensures the inference cost and the number of trainable parameters at test time remain identical between the two approaches. The performance improvement shown in the table is, therefore, not due to increased model capacity during inference but rather the effective use of unlabeled data for better representation learning.
>
> We did consider the idea of increasing the capacity of the supervised model by widening intermediate layers or adding more parameters, but our focus here was on comparing standard architectures with and without semi-supervised learning. We believe the strong results from our SSL approach demonstrate the benefit of leveraging unlabeled data, even without increasing the model’s capacity at inference. However, exploring capacity scaling for supervised models is indeed an interesting direction for future work.
>
> $\textbf{Q5:}$ "Is it possible to check if the two parallel models utilize complementary or less-correlated features? The aim of the feature discrepancy loss is that, however, is it possible to check if it serves the purpose? The feature discrepancy loss can be minimized by just having two shifted versions of similar feature vectors, which results from shuffling the order of the filters in the intermediate levels."
>
> To check whether the two parallel models utilize complementary or less-correlated features, we visualized the correlation between features of each sample among branches using cosine similarity. For more details, please refer to Section A in the Appendix of the revised paper.
>
> $\textbf{C: Concern; Q: Question}$

---

> ### Comment · Area_Chair_4Cpw · 2025-03-13
> **Rebuttal Response: Reviewer pe22**
>
> Dear reviewer pe22,
>
> Could you please review the author's rebuttal and submit your final response for the rebuttal. It's very important to have your final review and final decision.
>
> Best,
> ACT

---

### Official Review · Reviewer_7reP · 2025-02-20

**Confidence:** 5
**Preliminary Rating:** 5
**Recommendation:** Oral

**Summary:**

The paper presents a novel Dual Mask Ensemble (DME) approach for semi-supervised skin lesion segmentation. The method integrates a dual-branch co-training network with a feature discrepancy loss, aiming to improve feature diversity and reduce confirmation bias. The approach is evaluated on the ISIC-2018 and HAM10000 datasets, achieving state-of-the-art segmentation performance across various metrics.

**Strengths:**

The Dual Mask Ensemble (DME) strategy leverages both self-generated and opposing model predictions, improving the reliability of pseudo-labeling. Quantitative comparisons show that the proposed method outperforms existing semi-supervised approaches (PseudoSeg, CCT, CPS, UniMatch). The method is tested under different labeled/unlabeled data ratios, validating its effectiveness under limited supervision.

**Weaknesses:**

1. The model's real-world applicability in clinical settings is not extensively discussed. Future work could involve testing on real patient data or involving dermatologists in evaluation.
2. While the ablation study explores different mask refinement strategies, a deeper analysis of the contribution of Feature Discrepancy Loss (beyond Dice/Jaccard improvements) would strengthen the argument for its inclusion.

**Detailed Comments:**

NA

**Justification Of The Preliminary Rating:**

The paper presents a well-structured and technically sound approach to semi-supervised skin lesion segmentation. The Dual Mask Ensemble with Feature Discrepancy Loss introduces novel contributions to the field, improving segmentation reliability and robustness. While clinical validation and scalability remain open questions, the proposed method shows strong promise for improving AI-driven dermatological analysis.

**Questions To Address In The Rebuttal:**

1. How does your method handle challenging cases, such as rare skin lesion types or images with significant artifacts (e.g., hair, poor lighting, occlusions)?
2. Can you provide a deeper analysis of the impact of Feature Discrepancy Loss (FDL)?
3. How does your approach perform on out-of-distribution samples or datasets beyond ISIC-2018 and HAM10000?

---

> ### Author Response · Authors · 2025-03-08
>
> We truly appreciate the reviewer’s careful evaluation and insightful comments, which have guided us in refining our paper. We would like to take this opportunity to address the reviewer’s concerns, resolve any points of confusion, and improve our paper's clarity.
>
> $\textbf{Q1:}$ "How does your method handle challenging cases, such as rare skin lesion types or images with significant artifacts (e.g., hair, poor lighting, occlusions)?"
>
> Our method demonstrates robust performance even on challenging cases, including rare skin lesion types and images with significant artifacts such as hair occlusions, poor lighting, and complex backgrounds. You can see the figure on the revision paper.
>
> $\textbf{Q2:}$ "Can you provide a deeper analysis of the impact of Feature Discrepancy Loss (FDL)?"
>
> The Feature Discrepancy Loss (FDL) is essential in our framework, ensuring that the two sub-networks learn diverse and complementary representations. Without FDL, there’s a risk that both networks collapse into similar feature spaces, limiting their ability to provide distinct views on the data. By enforcing feature-level divergence, FDL improves the quality of combined mask predictions and enhances the robustness of pseudo-labeling, especially on challenging cases. Our ablation study, in Table 4, shows that adding FDL consistently boosts performance across metrics like Dice and Jaccard, confirming its critical role in the overall effectiveness of the model.
>
> $\textbf{Q3:}$ "How does your approach perform on out-of-distribution samples or datasets beyond ISIC-2018 and HAM10000?"
>
> Instead of evaluating on completely out-of-distribution samples, we perform cross-domain evaluation by training on one dataset and evaluating on the other. Specifically, we use the best-performing model weights from each dataset (ISIC2018 and HAM10000) and test them on the other dataset. The mentioned visualization on the full version of the revision paper.
>
> $\textbf{S4:}$ "A deeper analysis of the contribution of Feature Discrepancy Loss (beyond Dice/Jaccard improvements) would strengthen the argument for its inclusion."
>
> Thanks to the reviewer’s suggestion, we have added a more in-depth analysis and additional visualizations of the Feature Discrepancy module in the Appendix of the revised paper. For more details, please refer to Section A in the Appendix.
>
> $\textbf{Q: Question; }$
> $\textbf{S: Suggestion}$

---

> ### Comment · Area_Chair_4Cpw · 2025-03-13
> **Rebuttal Response: Reviewer 7reP**
>
> Dear reviewer 7reP,
>
> Could you please review the author's rebuttal and submit your final response for the rebuttal. It's very important to have your final review and final decision.
>
> Best,
> ACT

---

### Official Review · Reviewer_k9c2 · 2025-02-22

**Confidence:** 4
**Preliminary Rating:** 2
**Final Rating:** 3

**Summary:**

This paper proposes a semi-supervised medical image segmentation method featuring two new designs. The first is a Dual Mask Ensemble (DME) module, which combines predictions from two models trained on weakly augmented inputs to enhance the quality of pseudo-labels. The second is a feature discrepancy loss applied at the bottleneck between the two models to encourage disagreement. The effectiveness of the proposed method was evaluated on two publicly available skin lesion datasets.

**Strengths:**

1. The study addresses semi-supervised skin lesion segmentation, a topic of significant importance to the medical AI community.
2. The paper is well-structured and clearly written, making it easy to follow and understand.

**Weaknesses:**

1. Marginal innovation. The DME module essentially serves as a data augmentation strategy, and the concept of feature discrepancy between sub-networks has been widely explored in existing studies, such as [A, B]. Could the authors clarify what new insights or advancements their approach provides compared to these similar works?
2. Limited performance gain. Compared to UniMatch, the proposed method shows less than a 1% improvement in Dice score. Additionally, comparisons with more recent SSL approaches are crucial to fully demonstrate the effectiveness of the proposed method.
3. The evaluation is limited to 2D datasets; extending the proposed method to 3D datasets with other modalities is necessary, as no specific design has been made for skin lesions.

[A] Conflict-Based Cross-View Consistency for Semi-Supervised Semantic Segmentation. CVPR 2023.

[B] Consistency-guided Differential Decoding for Enhancing Semi-supervised Medical Image Segmentation. IEEE-TMI 2024.

**Detailed Comments:**

I appreciate the efforts made by the authors in the development of this work. However, the proposed approach does not offer significant new insights to the SSL community. Additionally, a more thorough review of recent advancements in SSL is needed to contextualize the method and highlight its contributions more effectively.

**Justification Of The Final Rating:**

The rebuttal effectively addresses my primary concerns. However, I would still like to see experimental evidence demonstrating that the feature-level regularization surpasses the differential decoding approach in [B]. Additionally, evaluating the generalizability of this method across a broader range of medical datasets would further strengthen its impact in the future. Considering these points, I will raise my score to borderline.

**Justification Of The Preliminary Rating:**

The work lacks novelty, as similar techniques have been extensively explored in prior works [A, B], with no clear justification of how this approach significantly advances beyond them. The performance gain is minimal, with less than a 1% Dice score improvement over UniMatch, and comparisons with more recent SSL methods are missing. Given the marginal innovation, limited performance improvement, and narrow experimental scope, the paper’s contributions are insufficient for acceptance.

**Questions To Address In The Rebuttal:**

refer to Weaknesses and Comments

---

> ### Author Response · Authors · 2025-03-08
>
> We sincerely thank the reviewer for their thoughtful and constructive feedback, which has helped us improve the clarity and quality of our work. Below, we address the reviewer’s concerns and clarify any misunderstandings.
>
> $\textbf{Q1:}$ "Marginal innovation. The DME module essentially serves as a data augmentation strategy, and the concept of feature discrepancy between sub-networks has been widely explored in existing studies, such as [A, B]. Could the authors clarify what new insights or advancements their approach provides compared to these similar works?"
>
> Although our works have some similar design objectives with [A] and [B], we introduced two key innovations compared to those references:
>
> First, while the Dual Mask Ensemble (DME) module can be viewed as a type of ensembling technique, it goes beyond standard strategies by adaptively combining predictions from dual sub-networks based on their confidence and consistency, which leads to more stable pseudo-labels and better generalization, addressing the issue of unreliable predictions often observed in semi-supervised learning. In contrast, [A] relies on conflict-based consistency but lacks an explicit mechanism to resolve low-confidence predictions. In Table 5, our loss design achieved a clear improvement compared to CCVC in most metrics. To clarify the effectiveness of our method compared to [A], we visualized the feature correlation among samples from both branches using cosine similarity. For more details, please refer to Section A in the Appendix of the revised paper.
>
> Second, the Feature Discrepancy Loss (FDL) in our framework is designed to align feature representations between sub-networks, enhancing the diversity and complementarity of their predictions. This feature-level regularization goes beyond the differential decoding approach in [B], which primarily focuses on decoder-level discrepancy learning without addressing representation-level discrepancies.
>
> These contributions together offer a more robust and effective co-training mechanism, leading to significant improvements in semi-supervised segmentation performance.
>
> $\textbf{C2:}$ "Limited performance gain. Compared to UniMatch, the proposed method shows less than a 1\% improvement in Dice score. Additionally, comparisons with more recent SSL approaches are crucial to fully demonstrate the effectiveness of the proposed method."
>
> It is true that we did not achieve impressive results from the proposed method compared to UniMatch, especially on 8\% setting of ISIC-2018. However, our extended experimental setting on ISIC-2018 reveals a more promising performance - the smaller the amount of labeled samples, the more robust we achieved with our method. Please refer to the main ISIC-2018 quantitative table.
>
> $\textbf{C3:}$ "The evaluation is limited to 2D datasets; extending the proposed method to 3D datasets with other modalities is necessary, as no specific design has been made for skin lesions."
>
> We agree that the technical design of our method can be utilized not only on 2D skin but also be effective on another anatomy and image modality, such as 3D. However, our work is designed for
> skin lesion segmentation medical tasks under a real-world scenario, where there is a huge amount of unlabeled dermatology data but a lack of labeled samples. Our works aim to provide technical insight in a semi-supervised setting, specifically for real-world dermatology diagnosis, which we believe matches the MIDL 2025 objective. Having said that, we appreciate suggestions, and we would extend the work further across modalities to validate the model generalization in a general medical image segmentation setting.
>
> $\textbf{Q: Question; C: Concern}$
>
> $\textbf{References:}$
>
> [A] Conflict-Based Cross-View Consistency for Semi-Supervised Semantic Segmentation. CVPR 2023.
>
> [B] Consistency-guided Differential Decoding for Enhancing Semi-supervised Medical Image Segmentation. IEEE-TMI 2024.

---

> ### Comment · Area_Chair_4Cpw · 2025-03-13
> **Rebuttal Review: Reviewer k9c2**
>
> Dear reviewer k92c,
>
> Could you please review the author's rebuttal and submit your final response for the rebuttal. It's very important to have your final review and final decision.
>
> Best,
> ACT

---

### Author Rebuttal · Authors · 2025-03-08

**Rebuttal:**

Dear reviewers, We appreciate your detailed reviews and concrete suggestions for improvement. We are especially grateful for reviewers' mentioning that our paper is "well-structured," "clearly written," "well justified," and "the proposed method shows strong promise for improving AI-driven dermatological analysis."

In our global response, we highlight the main changes we made as follows:

$\textbf{1. Extended quantitative and qualitative experiment results}$

- To address concerns about the performance gap among methods in ISIC-2018, we have conducted extra experiments on 1\% setting on ISIC-2018 dataset. Please refer to Table 1.
- We added Figure 5 in the Appendix to demonstrate robust performance even on challenging cases.
- We added Figure 6 in the Appendix to visualize performing cross-dataset validation for out-of-distribution performance evaluation.

$\textbf{2. Refined the writing and added visualization figure}$

- Thanks to reviewers' comments, we have refined our introduction part with many extra information. We first rewrote to correct the opening sentence about the importance of lesion segmentation with reference.
- We have added more related works about recent advancements in SSL to contextualize and highlight our contributions more effectively. We have also rewritten and highlighted the improvement of our methods compared to [1] and [2].
- To clarify our framework contribution thoroughly, we have added Figure 2, which demonstrates each module from a feature perspective.

$\textbf{3. Conducted deeper analysis of FDL}$

 - We added Table 5 in section 3.4.3 to make our design loss robustness visible. Although our works have some design objectives similar to those of [1] and [2], we updated two key innovations compared to those references in the revision paper. In Table 5, our loss design achieved a clear improvement compared to CCVC in most metrics.
- We visualized the feature correlation among samples from both branches using cosine similarity. For more details, please refer to Appendix A.
- Besides, our FDL aligns feature representations across sub-networks, enhancing prediction diversity and complementarity. Unlike [2], which targets decoder-level discrepancies, FDL regularizes at the feature level.

$\textbf{References}$

[1] Conflict-Based Cross-View Consistency for Semi-Supervised Semantic Segmentation. CVPR 2023.

[2] Consistency-guided Differential Decoding for Enhancing Semi-supervised Medical Image Segmentation. IEEE-TMI 2024.

**Supporting Material:**

/attachment/e087e3b4c80dde707483436b997deac4c29c2707.pdf

---

### Meta-Review · Area_Chair_4Cpw · 2025-03-21

**Recommendation:** Reject
**Confidence:** 3

**Metareview:**

This paper proposes a semi-supervised medical image segmentation method for skin lesion detection, combining a Dual Mask Ensemble (DME) module and a feature discrepancy loss. While the paper addresses a relevant problem and is clearly written, the reviewers raised significant concerns that were not adequately addressed in the rebuttal, leading to a recommendation for rejection. The primary concern is the marginal innovation of the proposed method. The DME module is essentially a data augmentation strategy, and the concept of feature discrepancy is well-explored in existing literature. The authors' attempts to differentiate their approach from previous work were unconvincing, failing to demonstrate a substantial advancement in the field.

Furthermore, the performance gains reported are minimal, particularly on the ISIC dataset, raising doubts about the method's practical significance. The authors' explanation that the method performs better with less labeled data does not compensate for the lack of substantial improvements in overall performance. The limitation to 2D datasets also restricts the method's applicability and generalizability. While the rebuttal addressed some technical questions, it did not resolve the fundamental issue of limited novelty and marginal performance gains. The authors' claims of robustness and improvement over existing methods were not sufficiently supported by experimental evidence or in-depth analysis.

Given the lack of significant innovation, the marginal performance improvements, and the limited scope of the evaluation, the paper does not meet the standards for acceptance. The rebuttal did not provide compelling evidence to overcome these weaknesses. Therefore, I recommend rejecting the paper.